# Identification of Shared Neoantigens in *BRCA1*-Related Breast Cancer

**DOI:** 10.3390/vaccines10101597

**Published:** 2022-09-22

**Authors:** Lucksica Ruangapirom, Nannapat Sutivijit, Chinachote Teerapakpinyo, Apiwat Mutirangura, Chatchanan Doungkamchan

**Affiliations:** 1Department of Anatomy, Faculty of Medicine, Chulalongkorn University, Bangkok 10330, Thailand; 2Doctor of Medicine Program, Faculty of Medicine, Chulalongkorn University, Bangkok 10330, Thailand; 3Department of Medicine, King Chulalongkorn Memorial Hospital, Bangkok 10330, Thailand

**Keywords:** cancer vaccine, *BRCA1*, breast cancer, shared neoantigens

## Abstract

Personalized neoantigen-based cancer vaccines have been shown to be safe and immunogenic in cancer patients; however, the manufacturing process can be costly and bring about delays in treatment. Using off-the-shelf cancer vaccines targeting shared neoantigens may circumvent these problems. Unique mutational signatures and similar phenotypes found among *BRCA1*-mutated breast cancer make it an ideal candidate for discovering shared neoantigens within the group. We obtained genome sequencing data of breast cancer samples with or without somatic *BRCA1* mutations (*BRCA1*-positive and *BRCA1*-negative, respectively) from the three public cancer databases; The Cancer Genome Atlas (TCGA), International Cancer Genome Consortium (ICGC), and Catalogue of Somatic Mutations in Cancer (COSMIC); and from three studies with whole genome/exome sequencing data of samples with germline *BRCA1* mutations. Data were analyzed separately within the same database/cohort. We found *PIK3CA* H1047R, E545K, E542K, and N345K recurrently in *BRCA1*-negative groups across all databases, whereas recurrent somatic mutations in *BRCA1*-positive groups were discordant among databases. For germline *BRCA1*-mutated breast cancer, *TP53* R175H was unanimously the most frequent mutation among the three germline cohorts. Our study provides lists of potential shared neoantigens among *BRCA1*-related breast cancer, which may be used in developing off-the-shelf neoantigen-based vaccines.

## 1. Introduction

The cancer vaccine is an approach to cancer immunotherapies involving the activation of T cells against tumor antigens and can be used as a therapeutic or preventive measure for cancer treatment [1]. Antigen targets for cancer vaccines are (1) over-expressed self-antigens known as tumor-associated antigens (TAAs); (2) cancer/testis antigens, which are exclusively expressed in reproductive tissues; and (3) neoantigens, which are exclusively expressed in tumors [2,3]. Therefore, the ideal target for cancer vaccines is neoantigens because their absence in normal tissue mitigates the chance of autoimmune attack [4]. Neoantigen-based cancer vaccine manufacture begins with the identification of somatic mutations by comparing the tumor genome to normal tissues within the same individuals. Somatic mutations that are found uniquely in tumors are then assessed for antigenicity by computerized algorithms and are validated in vitro/in vivo before being administered to patients [5,6,7]. Although the personalized cancer vaccine approach has been shown to elicit immune responses in various cancers, complex and completely individualized manufacturing processes result in high treatment cost and delayed treatment availability, preventing treatment access for all patients [8]. To overcome this problem, off-the-shelf cancer vaccines targeting neoantigens that are “common” or “shared” among groups of cancer patients may help reduce the cost and time of access to the cancer vaccine. In this study, we aimed to identify shared (or common or public) neoantigens found recurrently in *BRCA1*-related breast cancer patients that may be used as target neoantigens for shared vaccine development.

Breast cancer is the leading cause of death by cancer among women (approximately 700,00 deaths; 15.5% of total women cancer deaths) and the most commonly diagnosed cancer among all cancers and sexes, with an estimated 2.3 million new cases in 2020 (11.7% of total cases) [9]. Breast cancer affects women worldwide with similar incidence and mortality rates [9]. Approximately 1–4% of all breast cancer cases are *BRCA1*-related [10,11]. We chose to identify shared neoantigens in *BRCA1*-related breast cancer because of its characteristic mutational signatures suggesting a pattern in mutational events within the group [12,13]. Additionally, *BRCA1*-mutated breast cancer also shares phenotypic similarities such as morphology, molecular subtype, and responsiveness to poly-(adenosine diphosphate-ribose) polymerase inhibitors (PARPi) treatment. To further illustrate, studies have found approximately 70% of *BRCA1*-mutated breast cancer to exhibit a basal-like pattern in molecular subtype compared to 20% in *BRCA1*-wild-type breast cancer; 57–68% of *BRCA1*-mutated breast cancer exhibit triple-negative breast cancer (TNBC) in surrogate subtype compared to 13% in *BRCA1*-wild-type breast cancer, and a 50–79% response to PARPi compared to 10–33% in *BRCA1*-wild-type cases [14,15,16,17,18]. Because of these similarities within *BRCA1*-mutated breast cancer, we hypothesized that some neoantigens may be found recurrently across individuals with *BRCA1* mutations and may be used as neoantigens for off-the-shelf cancer vaccines, both for therapeutic purposes for cases with somatic *BRCA1* mutations, and for preventive purposes for those with germline *BRCA1* mutations.

The concept of shared-antigen cancer vaccines has been investigated, especially during the past decade [19,20]. Recent studies have shown successful treatments using neoantigen-based shared cancer vaccines in some types of cancers such as *IDH1* R132H for glioblastoma and *KRAS* G12D for colon cancer [21,22,23]. Other common neoantigens that have been identified and proposed as targets for cancer vaccines are *TP53* R175H and *PIK3CA* H1047R for gastric cancer, *RET* M918T for thyroid cancer, and common frameshift mutations for microsatellite instability-high (MSI-H) tumors [24,25,26,27]. Nevertheless, shared target antigens reported in breast cancer, such as *HER2* and *MUC1*, are TAAs, not neoantigens [28,29,30,31]. A study, which recently reported shared neoantigen targets in breast cancer such as *PIK3CA* H1047R E545K N345K and *AKT1* E17K, was conducted on unspecified breast cancer samples [32]. To the best of our knowledge, shared neoantigen targets for *BRCA1*-related breast cancer have not been reported.

In this study, we proposed potential neoantigen targets that are found commonly in *BRCA1*-positive, -negative, and germline *BRCA1*-mutated samples. We included samples from large open-access public cancer genome databases: TCGA, ICGC, and COSMIC to identify top recurrent mutations from which we also predicted antigenicity of encoded proteins. Our study provided lists of potential shared neoantigens among *BRCA1*-related breast cancer, which may be used for developing off-the-shelf neoantigen-based vaccines, and may reflect different mutational consequences among somatic, germline *BRCA1*-mutated, and *BRCA1*-wild-type breast cancer that should be further investigated.

## 2. Materials and Methods

### 2.1. Sample Identification

We searched 3 cancer genome databases: The Cancer Genome Atlas (TCGA), the International Cancer Genome Consortium (ICGC), and the Catalogue of Somatic Mutations in Cancer (COSMIC) for sequencing data of breast cancer samples with and without the *BRCA1* mutation. Data related to samples from these 3 databases were collected with original informed consent. For TCGA and ICGC databases, the search term “primary site is breast and gene is *BRCA1*” was used to locate samples with sequencing data on the *BRCA1* gene. We found 154 samples on TCGA (data release version: 3 May 2022) and 1852 samples on ICGC databases (data release version: 27 March 2019). Among these samples, 27/154 (17.53%) from TCGA and 106/1852 (5.72%) from ICGC were shown to harbor *BRCA1* mutations. For the COSMIC database (data version: 28 May 2021), we found 245 samples using the search terms “gene: *BRCA1*, tissue: breast”. Among the samples with *BRCA1* mutations, we only included samples with “pathogenic” or “likely pathogenic” *BRCA1* somatic mutations by the criteria of the American College of Medical Genetics and Genomics (ACMG) 2015 in our study [33]. Exclusion criteria were (1) samples with single gene target sequencing, and (2) samples with known germline *BRCA1* mutations reported by previous studies [12,34,35] (Appendix A). The samples identified by previous studies as germline *BRCA1* mutations were to be analyzed separately. With these criteria, we were able to include 12, 15, and 66 samples from TCGA, ICGC, and COSMIC databases, respectively, into our study. These samples with “pathogenic” or “likely pathogenic” somatic *BRCA1* mutations will be referred to in this paper as “*BRCA1*-positive”. Samples that were known to have wild-type *BRCA1* sequence were classified as “*BRCA1*-negative”. All sample IDs included in this study can be found in Appendix A for germline *BRCA1*-mutated samples and Appendix A for *BRCA1*-positive and -negative samples.

### 2.2. Data Analyses

For TCGA and ICGC databases, *BRCA1*-positive and -negative samples were grouped into separate folders using web interfaces. Then, data on all mutations of samples in each group were downloaded onto a local computer for analysis. For the COSMIC database, all mutation data were downloaded from the “All mutations in census gene” and “Non-coding variant” sections. All mutations were assessed for variant type, single nucleotide substitution classification, coding-region variant classification, and variant count per sample by simple counting. Data of top recurrent mutated genes and top recurrent somatic mutations were obtained by counting and were also available via the TCGA and ICGC web interfaces. Top recurrent somatic mutations on coding regions from all databases were also assessed by simple counting. 

### 2.3. Antigenicity Prediction of Recurrent Somatic Mutations

We assessed antigenic potentials of the top recurrent somatic mutations by calculating binding affinity between mutated epitopes and Major Histocompatibility Complex (MHC) class I. Binding affinities were calculated using NetMHCpan and The Immune Epitope Database (IEDB) algorithms [36,37,38]. We used 2 prediction methods, NetMHCpan BA 4.1 and NetMHCpan EL4.1. MHC Class I/peptide pairs with stronger than moderate binding affinity (IC50 < 500) will be determined as possibly antigenic. Such mutations will be considered antigenic only when neoepitopes are available. We included 145 MHC class I alleles in this study to cover 99% of all MHC class I in the worldwide population [38]. 

Binding affinities between neoepitopes and MHC class II were calculated using the recommended method by the Immune Epitope Database (IEDB) algorithms. If none of these methods were available for the allele, NetMHCIIpan 4.0 was used. Twenty-seven MHC class II alleles were used in this study to cover 99% of all MHC class II in the worldwide population [39]. We selected 12-mer to 18-mer peptides. The IEDB recommended that selections were based on a consensus percentile rank of the top 10%. Alternatively, peptides with binding affinity to MHC Class II at less than 1000 nM were classified into binders. Antigenic epitopes were generated exclusively from mutated proteins and cannot be found in wild-type proteins. 

### 2.4. Allele and Haplotype Frequency Calculations

Average allele frequency of each MHC class I was calculated using data from the Allele Frequency Net Database (https://www.allelefrequencies.net) (access date: 1 February 2022) [40]. The data inclusion criteria for allele frequency calculation were set at “all population, all countries, all sources of the dataset, all regions, all ethnicities, all type of study, gold population standard only”, with gold qualities being defined as (1) having allele frequency totaled or close to 1, (2) the sample size of more than 50, and (3) the frequencies of four-digit resolution. Average haplotype frequencies were calculated using data also from the Allele Frequency Net database. The data inclusion criteria for haplotype frequency calculation were set at “all population, all countries, all sources of the dataset, all regions, all ethnicities, all type of study, sort by haplotype, 2 loci test”.

### 2.5. Statistical Analysis

Mann–Whitney U test was performed to compare variant counts between *BRCA1*-positive and -negative samples within the same databases using IBM SPSS Statistics for Windows, Version 22.0. Armonk, NY, USA: IBM Corp. *p* < 0.05 was considered statistically significant.

## 3. Results

### 3.1. Characteristics of BRCA1-Positive and BRCA1-Negative Samples

Characteristics of breast cancer samples with pathogenic or likely pathogenic *BRCA1* mutations (referred to as *BRCA1*-positive) are shown in Table 1. The mean age of samples in TCGA, ICGC, and COSMIC was 59.58 (±12.86), 58.73 (±16.26), and 54.14 (±14.91) years, respectively, excluding 38 samples with unknown age from COSMIC. All samples were from female donors except for one male donor from COSMIC. Most samples were stage 2 breast cancer in TCGA (8/12, 66.67%) and ICGC (9/15, 60.00%), but the unknown stage was the majority in COSMIC (52/66, 78.78%). For histology, the majority of samples were reported as infiltrating ductal carcinoma in TCGA (11/12, 91.67%) and ICGC (14/15, 93.33%), and unknown histology (42/66, 63.63%) followed by infiltrating ductal carcinoma (20/66, 30.30%) in COSMIC. For molecular subtype, 29/66 (43.94%) of COSMIC samples were hormonal-receptor positive, whereas most samples from TCGA and ICGC showed unidentified molecular subtype. For sequencing data type, all samples in TCGA were whole exome sequencing (WES) (12/12, 100%). Samples in ICGC were reported in WES (8/15, 53.33%) and Whole genome sequencing (WGS) (7/15, 46.67%). Samples in COSMIC consisted of WES (9/66, 13.64%), WGS (11/66, 16.67%), and target sequencing (46/66, 69.69%).

*BRCA1*-negative samples identified by samples with no mutations at the *BRCA1* gene were also included in our analysis for comparison. There were 123, 1714, and 3158 *BRCA1*-negative samples identified in TCGA, ICGC, and COSMIC, respectively (Table 2). The mean age of samples in TCGA, ICGC, and COSMIC was 58.34 (±13.68), 56.67 (±13.85), and 59.39 (±12.49) years, respectively, with 3 samples from TCGA, 116 samples from ICGC, and 2953 samples from COSMIC with unknown age. Female donors contributed 119 samples (96.74%) from TCGA, 1699 (99.12%) from ICGC, and 2989 (94.65%) samples from COSMIC. For the stage at which samples were obtained, most samples in TCGA were from stage 2 (65/123, 52.84%), while most had unknown staging in ICGC (1532/1714, 89.38%) and COSMIC (2244/3158, 71.06%). For histology, infiltrating ductal carcinoma was the majority in TCGA (102/123, 82.92%). Most samples in ICGC (1144/1714, 66.74%) and COSMIC (1491/3158, 47.21%) had unknown histology, followed by infiltrating ductal carcinoma (485/1714, 28.29%) in ICGC and (1304/3158, 41.29%) in COSMIC. For molecular subtype, the information was not provided for most samples in the databases. However, ER-positive/HER2-negative and hormone receptor-positive were the major subtypes in ICGC (477/1714, 27.82%) and COSMIC (1132/3158, 35.85%), respectively, if information was provided. For the sequencing data type, all samples in TCGA were whole exome sequencing (WES) (123/123, 100%). Samples in ICGC were reported in WES (1035/1714, 60/38%) and Whole genome sequencing (WGS) (679/1714, 49.61%). Most samples in COSMIC were WES (9/3158, 0.28%), WGS (70/3158, 2.22%), and mostly target sequencing (3079/3158, 97.50%). The aforementioned data showed similar sample demographics between *BRCA1*-positive and *BRCA1*-negative samples within the same database.

### 3.2. Mutational Landscapes of BRCA1-Positive and -Negative Samples

To compare variant types between *BRCA1*-positive and -negative samples, we classified all somatic mutations, from both coding and non-coding regions, into single nucleotide variation (SNV), deletion (DEL), insertion (INS), and others. “Others” is comprised of mutations that could not be classified into any of the previous categories. Insertion and deletion sizes from ICGC were less than 200 bp. We found that SNV was the majority of *BRCA1*-positive (TCGA 1475/1558, 94.67%; ICGC 92,951/97,282, 95.54%; and COSMIC 27,057/28,643, 94.46%) and *BRCA1*-negative samples (TCGA 12,894/13,749, 93.78%; ICGC 2,569,039/2,703,478, 95.03%; and COSMIC 9445/12,214, 77.31%) in all databases (Figure 1A,B). Among these SNVs, C>T was found most abundant in both *BRCA1*-positive (TCGA 539/1475, 36.54%; ICGC 32,619/92,951, 35.09%; and COSMIC 7059/27,057, 26.08%) and *BRCA1*-negative samples (TCGA 6129/12,894, 47.53%; ICGC 844,924/2,569,039, 32.88%; and COSMIC 4008/9445, 42.43%) (Figure 2A,B). The second most abundant SNV was C>G in *BRCA1*-positive samples from all databases. (TCGA 358/1475, 24.27%; ICGC 29,438/92,951, 31.65%; and COSMIC 5641/27,057, 20.84%) (Figure 2A). However, the second most abundant SNV in *BRCA1*-negative samples were varied among databases with C>G, C>A, and T>C being the second most abundant SNVs in TCGA (3807/12,894, 29.52%), ICGC (543,067/2,569,039, 21.13%), and COSMIC (1725/9445, 18.26%), respectively (Figure 2B). This showed that SNV was the major variant type with C>T mutations being the most abundant SNV in both *BRCA1*-positive and -negative samples.

Next, we classified the mutations at coding regions into seven types: missense, synonymous, nonsense, frameshift deletion, in-frame deletion, frameshift insertion, and in-frame insertion. We found that missense mutation was the most abundant mutation found at coding regions in both *BRCA1*-positive (TCGA 993/1558, 63.86%; ICGC 1797/97,282, 1.85%; and COSMIC 258/28,643, 0.90%) and *BRCA1*-negative samples (TCGA 8528/13,749, 62.03%; ICGC 88,553/2,703,478, 3.28%; and COSMIC 6804/12,214, 55.71%) in all databases, followed by synonymous mutations in TCGA and ICGC (Figure 3A,B). However, COSMIC databases showed frameshift deletions to be the second most abundant mutation in both *BRCA1*-positive and *BRCA1*-negative samples (Figure 3A,B). Taken together, a missense mutation is the most abundant variant type in coding regions in both *BRCA1*-positive and -negative samples.

To assess the tumor mutational burden of *BRCA1*-positive and -negative samples, we compared coding-region non-synonymous variant count per sample from each group within the same database. We excluded samples that were analyzed by target sequencing in the COSMIC database. Therefore, 20 *BRCA1*-positive and 79 *BRCA1*-negative from COSMIC were analyzed. We found that the median of total variant counts in *BRCA1*-positive samples was 83.50 (Q1 59.75-Q3 140.25), 95.00 (Q1 66.00-Q3 140.00), and 6.50 (Q1 4.00-Q3 8.00) in TCGA, ICGC, and COSMIC databases, respectively (Figure 4A). For *BRCA1*-negative samples, the median of variant counts in TCGA, ICGC, and COSMIC databases was 39.00 (Q1 22.00-Q3 65.00), 36.00 (Q1 23.00-Q3 62.00), and 2.00 (Q1 1.00-Q3 2.00), respectively (Figure 4B). We found significant differences in total non-synonymous variant counts between *BRCA1*-positive and *BRCA1*-negative samples in all databases (2-tailed *p*-value: <0.001 in TCGA, <0.001 in ICGC, and 0.001 in COSMIC) (Figure 4C). We also found significant differences in both SNV and indel counts between *BRCA1*-positive and -negative groups in all databases (*p*-value < 0.001, <0.001, and <0.001 in TCGA, ICGC, and COSMIC for SNV count and 0.019, 0.032, and 0.031 in TCGA, ICGC, and COSMIC for the indel count) (Appendix A).

### 3.3. Frequently Mutated Genes in BRCA1-Positive and -Negative Breast Cancer Samples

We found that *TP53* was the most frequently mutated gene in *BRCA1*-positive samples and accounted for 75.00% (9/12), 60% (9/15), and 59.09% (39/66) in TCGA, ICGC, and COSMIC, respectively (Figure 5A). *TTN* was the other top mutated gene in ICGC (9/15, 60%) and was the second most mutated gene in TCGA (6/12, 50.00%) but was not reported among the top mutated genes in COSMIC. *PIK3CA*, which was one of the top mutated genes in *BRCA1*-negative samples, was found to be 16.67% (2/12) of *BRCA1*-positive samples in TCGA, 13.33% (2/15) in ICGC, and 15.15% (10/66) in COSMIC, yet was not among the top mutated genes found in *BRCA1*-positive samples. For *BRCA1*-negative samples, *TP53* and *PIK3CA* were the most frequently mutated genes in TCGA (*TP53*: 46/123, 37.39%; *PIK3CA*: 37/123, 30.08%) and COSMIC (*TP53*: 955/3158, 30.24%; *PIK3CA*: 1247/3158, 39.48%) and were also found in the top 10 most frequently mutated genes in ICGC (*TP53*: 555/1714, 32.38%; *PIK3CA*: 583/1714, 34.01%) (Figure 5B). *CSMD1* was the most frequently mutated gene in ICGC (627/1714, 36.58%) and was in the top 20 mutated genes in TCGA (7/123: 5.69%) but was not among the most frequently mutated genes in COSMIC (Figure 5B). To summarize, *TP53* and *TTN* were the most frequently mutated genes in *BRCA1*-positive samples, whereas *PIK3CA* and *TP53* were the most frequently mutated genes in *BRCA1*-negative samples; thus, this may reflect different the mutational pathways between *BRCA1*-positive and *BRCA1*-negative samples.

### 3.4. Recurrent Somatic Mutations in BRCA1-Positive and BRCA1-Negative Breast Cancer Samples

To propose candidate neoantigens for generalized breast cancer vaccine development, in both *BRCA1*-positive and *BRCA1*-negative samples, we looked at the top recurrent somatic mutations exclusively at coding regions (Figure 6A,B). We found missense *PIK3CA* H1047R, E545K, N345K, and E542K consistently across all databases in *BRCA1*-negative samples. *PIK3CA* H1047R was reported in *BRCA1*-negative samples in all databases (TCGA 15/123, 12.19%; ICGC 198/1714, 11.55%; and COSMIC 488/3158, 15.45%) but was reported at various prevalence in *BRCA1*-positive samples (TCGA: 2/12, 16.67%; ICGC: 0/15, 0%; and COSMIC: 3/66, 4.54%). *PIK3CA* E545K and *PIK3CA* N345K were identified in *BRCA1*-negative samples with comparable prevalence in all databases (E545K 5.69–7.63%; N345K 1.69–3.25%) but were rarely found in *BRCA1*-positive samples (E545K 0–1.51%; N345K 0%), whereas *PIK3CA* E542K was identified at comparable prevalence in both *BRCA1*-positive (4.54–6.67%) and -negative (2.43–4.78%) in all databases. We found no dominant recurrent mutations among *BRCA1*-positive samples; however, *PIK3CA* H1047R, *PIK3CA* E542K, *TP53* Y220C, and *TP53* R196* were found among the top recurrent mutations in more than two databases (Figure 6A). This showed that while the recurrent coding-region mutations are still inconclusive for *BRCA1*-positive samples, *PIK3CA* H1047R, E545K, N345K, and E542K were consistently identified across all databases for *BRCA1*-negative samples and, therefore, may be used as common target neoantigens for the *BRCA1*-negative breast cancer vaccine.

To investigate the overlap of the top 20 recurrent mutations between *BRCA1*-positive and -negative groups, we used a Venn diagram to visualize the overlap of somatic mutations between the two groups within the same database (Appendix A). The data showed that *PIK3CA* H1047R, *TP53* R196*, and *GATA3* D336Gfs*17 to be overlapped in both subgroups in TCGA; *PIK3CA* E542K in ICGC; and *PIK3CA* H1047R, E542K, E545K, and *GATA3* D336Gfs*17 in COSMIC. Although these mutations were shown to be overlapped by the Venn diagram, the recurrent rates can be different between the subgroups (Appendix A). These findings also confirm that varied neoantigen sets may be necessary to provide good coverage for subgroups with different *BRCA1* statuses. 

Targeting multiple neoantigens at a time may provide more coverage for a generalized breast cancer vaccine. Therefore, we also calculated the cumulative coverage of recurrent mutations in both types of samples (Figure 6C,D). In *BRCA1*-positive samples, we found that the top 5 somatic mutations as displayed in the graph can cover 41.66% of all samples and the top 12 somatic mutations to cover 83.33% of the sample in TCGA; the top 6 somatic mutations to cover 26.67% and the top 17 to cover 66.67% of the samples in ICGC; and lastly, the top 5 to cover 19.69% and the top 20 to cover 24.24% of the samples in COSMIC (Figure 6C). For *BRCA1*-negative samples, we found that the top 5 somatic mutations as displayed in the graph to cover 26.01% of all samples and the top 20 somatic mutations to cover 33.33% of the sample in TCGA; the top 5 somatic mutations to cover 25.37% and the top 20 to cover 36.34% of the samples in ICGC; and lastly, the top 5 to cover 33.91% and the top 20 to cover 45.18% of the samples in COSMIC (Figure 6D).

### 3.5. Recurrent Somatic Mutations in Germline BRCA1-Mutated Breast Cancer Samples

Next, we aimed to identify candidate neoantigens to be used as a preventive cancer vaccine for germline *BRCA1* carriers. We identified three studies that reported next-generation sequencing data on samples confirmed to be germline *BRCA1* mutations. Those studies were conducted by Nik-Zanial et al. (2016) [12], by Nones et al. (2019) [34], and by Inagaki-Kawata et al. (2020) [35]. The summary of sample characteristics is shown in Appendix A. Seventy-eight total samples were obtained and analyzed separately. We found that missense *TP53* R175H was consistently the most frequent somatic mutation in all studies and accounted for 6.45%, 11.53%, and 9.52% in the studies of Nik-Zanial et al. (2016), Nones et al. (2019), and Inagaki-Kawata et al. (2020), respectively (Figure 7). Interestingly, *TP53* R175H was not found among the top 20 somatic mutations in all *BRCA1*-positive and -negative studies of ICGC and TCGA. 

Comparing recurrent mutations from germline *BRCA1*-mutated samples to those identified in *BRCA1*-positive/-negative samples, we found that missense *PIK3CA* H1047R mutation, which is one of the most frequently recurring in both *BRCA1*-positive and *BRCA1*-negative studies, was rarely found in germline *BRCA1*-mutated samples (Nik-Zanial et al.: 0%; Nones et al.: 3.84%; and Inagaki-Kawata et al.: 4.76%). Additionally, other frequently found somatic mutations in *BRCA1*-negative studies were also rarely found in germline *BRCA1*-mutated studies: missense *PIK3CA* E545K (Nik-Zanial et al.: 3.22%; Nones et al.: 0%; and Inagaki-Kawata et al.: 0%), missense *PIK3CA* E542K (Nik-Zanial et al.: 6.45%; Nones et al.: 0%; and Inagaki-Kawata et al.: 0%), missense *PIK3CA* N345K (Nik-Zanial et al.: 0%; Nones et al.: 0%; and Inagaki-Kawata et al.: 0%) (Figure 7). This information may indicate unique mutational consequences among samples with germline *BRCA1* mutations, non-specific *BRCA1*-positive mutations, and no *BRCA1* mutations. 

### 3.6. Predicted Antigenicity of Top Recurrent Mutations

We investigated the antigenic potential of the peptides from top recurrent somatic mutations by calculating their epitopes’ binding affinities with MHC class I. The calculated binding affinities of top recurrent mutations were obtained from NetMHCpan BA 4.1 and NetMHCpan EL 4.1. Candidate recurrent somatic mutations are predicted to be antigenic when their binding affinity was lower than 500 nM (Table 3). We found that most recurrent mutations are predicted to be antigenic except for *TP53* R175H and *TP53* R196*. For binding affinities with MHC class II, we found that most recurrent mutations were predicted to be antigenic except for *PIK3CA* E542K and *TP53* R196*, which resulted in a premature stop codon (Appendix A). A combination of *PIK3CA* H1047R, E542K, E545K, and N345K can cover 10.75% (10/93) of *BRCA1*-positive samples and 27.50% (1374/4995) of *BRCA1*-negative samples. It is noteworthy that the combination of these recurrent neoantigens can cover minimal to no samples with germline *BRCA1* mutations.

## 4. Discussion

Personalized neoantigen-based cancer vaccines have revolutionized personalized medicine; however, its highly-individualized manufacturing process may hinder its accessibility by all. Shared neoantigen vaccines have certain advantages over personalized approaches in cases where time and resources are limited and patients have such aggressive diseases that they cannot afford delayed treatment. In this study, we identified recurrent somatic mutations and potential shared neoantigen candidates in *BRCA1*-positive, -negative, and germline *BRCA1*-mutated breast cancer. We analyzed mutation data of the breast cancer samples with and without “pathogenic” or “likely pathogenic” *BRCA1* mutation as determined by ACMG 2015 criteria (referred to as *BRCA1*-positive and -negative, respectively) that were available on three public cancer databases: TCGA, ICGC, and COSMIC). We also reported mutational landscapes and frequently mutated genes in *BRCA1*-positive and -negative samples.

Our findings on mutational landscapes correspond with previous studies. Our results showed that SNV is the most abundant variant type with C>T being the most abundant SNV type in both *BRCA1*-positive and -negative groups. This corresponds with the study results by Zhou and colleagues in 5991 unspecified breast cancer samples [32]. Mutational burden represented by total variant counts showed differences between *BRCA1*-positive and -negative among all databases with the *BRCA1*-positive group harboring more SNVs and indels compared to the *BRCA1*-negative group. These results also correspond with findings by Nolan and colleagues who analyzed the tumor mutational burden by WES and found a marked enrichment of missense and indel mutations in *BRCA1*-mutated triple-negative breast cancer compared to the non-*BRCA1*-mutated group [41]. 

*TP53* and *PIK3CA* were found mutated in 30.24–37.39% and 30.08–39.48% of all *BRCA1*-negative samples, respectively. These mutational frequencies are similar to the findings in unspecified breast cancer samples (*TP53* 37%; *PIK3CA* 38%) [32,42]. In the *BRCA1*-positive group, however, *TP53* was found to be the top mutated gene with much higher frequencies of 59.09–75.00% of all *BRCA1*-positive samples. This corresponds with a previous finding that loss-of-function *TP53* is required for efficient tumor development in targeted *BRCA*-null mice [43]. Interestingly, *PIK3CA* mutations, which were the top mutation in *BRCA1*-negative and unspecified breast cancer, were found in the lower frequencies of 13.33–16.67% and were not among the top mutated genes in the *BRCA1*-positive group. This also correlates with the frequency of *PIK3CA* mutations in TNBC (16%), which is lower than in the hormonal positive subgroups (HR+/HER2 (42%) and HER2+ (31%)) [42]. Taken together with the fact that most *BRCA1*-mutated samples are TNBC (57–68%) [15,16], and *TP53* and *PIK3CA* mutations are often mutually exclusive [32], the relationship between hormonal status (TNBC), *TP53*, *PIK3CA*, and *BRCA1* mutation status may need to be further explored.

We reported *PIK3CA* H1047R, *PIK3CA* E545K, *PIK3CA* E542K, and *PIK3CA* N345K to be the top recurrent somatic mutations in *BRCA1*-negative samples. The mutation list is highly similar to the finding in breast cancer of unspecified genotypes (*PIK3CA* H1047R E545K N345K, and *AKT1* E17K) [32]. On the other hand, the recurrent somatic mutations in *BRCA1*-positive samples (*PIK3CA* H1047R, *TP53* Y220K, *TP53* R196*, and *PIK3CA* E542K) may not be reproducible across all databases, possibly due to low sample counts. However, the germline *BRCA1*-mutated samples showed *TP53* R175H to be the only top recurrent mutation unanimously across all three cohorts, and they are not found among the top mutations in *BRCA1*-positive or -negative samples. Therefore, the same set of neoantigens that cover one *BRCA1* status may not cover the others. The different sets of recurrent somatic mutations among *BRCA1*-positive, -negative, and germline *BRCA1* samples reported in this study may reflect different mutational events among these groups. 

Most recurrent somatic mutations were predicted by NetMHCpan algorithms to be antigenic and their peptides can be presented by both MHC-class I and class II, except for *TP53* R175H; however, it was shown to be antigenic by the in vitro studies [44,45]. Other predicted neoantigens identified in this study such as *PIK3CA* H1047R were reported to elicit both CD4+ and CD8+ responses in vitro [46], whereas *PIK3CA* E545K and E542K were not found to be presented by MHC class I in vitro [46,47]. Therefore, in vitro validation of the predicted neoantigens reported by this study is necessary. The lack of in vitro testing is another limitation of this study in addition to the small sample size in the *BRCA1*-positive group.

This work provides a foundation for developing off-the-shelf neoantigen-based vaccines for *BRCA1*-related breast cancer, yet there are several critical steps prior to achieving the vaccine development goal. In our future study, we plan to validate the antigenicity of the candidate neoantigens in vitro by testing their ability to bind MHC molecules and their ability to elicit a T cell response. We plan to proceed to in vivo studies with the validated neoantigen candidates by using those shared neoantigens to immunize mouse models with *BRCA1*-mutated breast cancer. The vaccinated mice will be evaluated for antibody levels and T-cell activities against vaccinated neoantigens, and the clinical outcomes such as tumor size, progression, metastasis, and survival, will be compared to the unvaccinated control group.

## 5. Conclusions

Our study identified common somatic mutations that are predicted to be immunogenic in *BRCA1*-related breast cancer (*BRCA1*-positive, -negative, and germline *BRCA1* mutations). We reported *PIK3CA* H1047R, *PIK3CA* E545K, *PIK3CA* E542K, and *PIK3CA* N345K to be the top recurrent mutations in *BRCA1*-negative samples across all databases. On the other hand, *PIK3CA* H1047R, *TP53* Y220K, *TP53* R196*, and *PIK3CA* E542K, which were found recurrently in *BRCA1*-positive samples, were not consistent among the databases. *TP53* R175H was the top recurrent somatic mutation found uniquely in the germline *BRCA1*-mutated group. Collectively, our study provided lists of candidate neoantigens that may be used to develop off-the-shelf cancer vaccines for *BRCA1*-related breast cancer patients or as a preventive cancer vaccine in *BRCA1*-mutated carriers. However, in vitro validations of the candidates’ antigenicity and assessment of their ability to immunize and regulate cancer progression in vivo are the critical next steps, which we plan to include in our future study.

## Figures and Tables

**Figure 1 vaccines-10-01597-f001:**
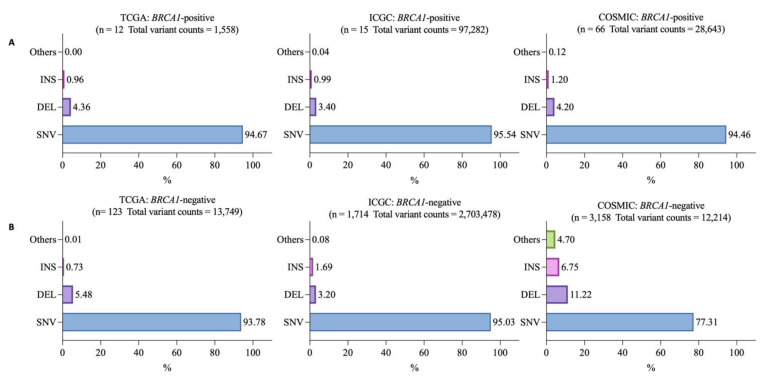
Percent variant type per total variant count in (**A**) *BRCA1*-positive and (**B**) *BRCA1*-negative breast cancer samples shown by each database. Numbers on the bar graphs indicate the percentage of variant type per total variant count. (INS = insertion, DEL = deletion, SNV = single nucleotide variant).

**Figure 2 vaccines-10-01597-f002:**
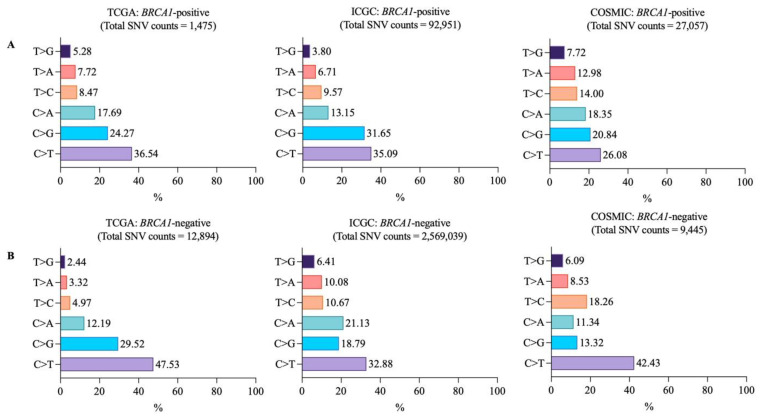
SNV classification of (**A**) *BRCA1*-positive and (**B**) *BRCA1*-negative breast cancer samples. Numbers on the bar graphs indicate the percentage of SNV per total SNV count.

**Figure 3 vaccines-10-01597-f003:**
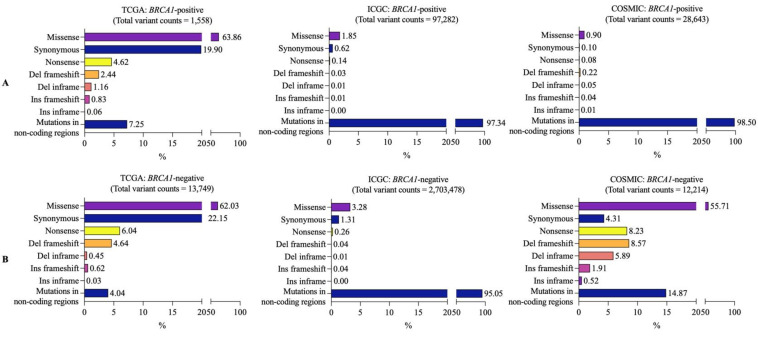
Coding-region variant classification in (**A**) *BRCA1*-positive and (**B**) *BRCA1*-negative breast cancer samples. Numbers on the bar graphs indicate the percentage of coding-region variant types per total variant count. (Del frameshift = frameshift deletion, Del inframe = inframe deletion, Ins frameshift = frameshift insertion, Ins inframe = inframe insertion).

**Figure 4 vaccines-10-01597-f004:**
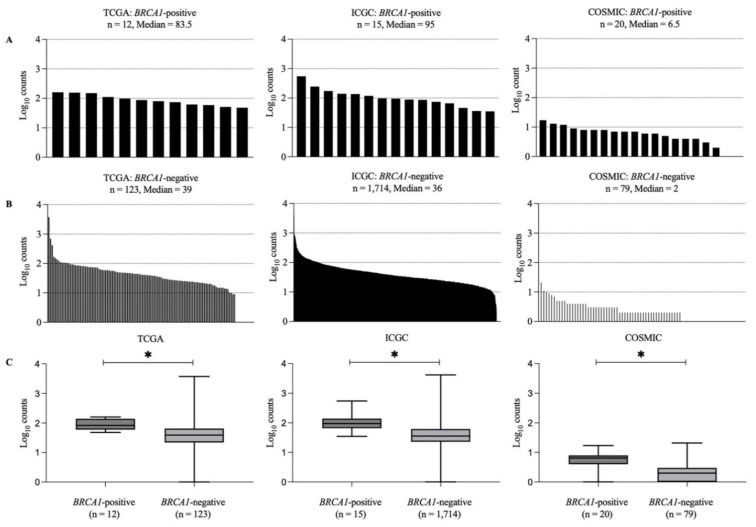
Total count of non-synonymous variants per sample of (**A**) *BRCA1*-positive and (**B**) *BRCA1*-negative breast cancer samples. Comparison of total count of non-synonymous variant per sample between *BRCA1*-positive and -negative within the same database using Mann–Whitney U test (**C**). The box plot showed median, the lower quartile, and the upper quartile. * Indicates *p* < 0.05. The *y*-axis represents log_10_ (variant counts of each sample). The number of samples is indicated below the box plot (*n*).

**Figure 5 vaccines-10-01597-f005:**
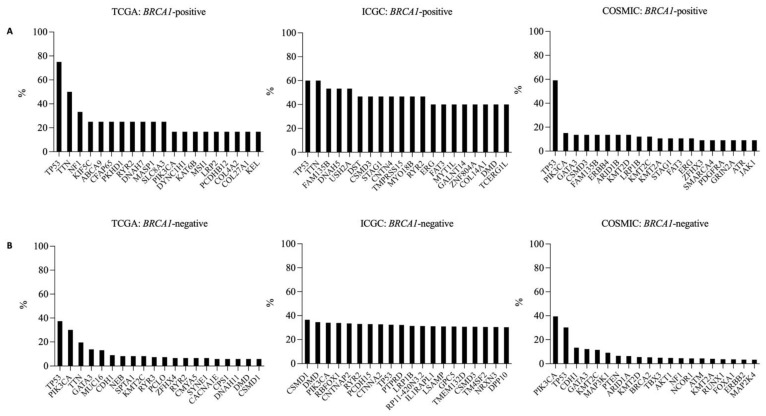
Top mutated genes in (**A**) *BRCA1*-positive and (**B**) *BRCA1*-negative breast cancer samples. *x*-axes indicate mutated genes and the *y*-axes indicate the percentage of all samples with the corresponding mutated gene.

**Figure 6 vaccines-10-01597-f006:**
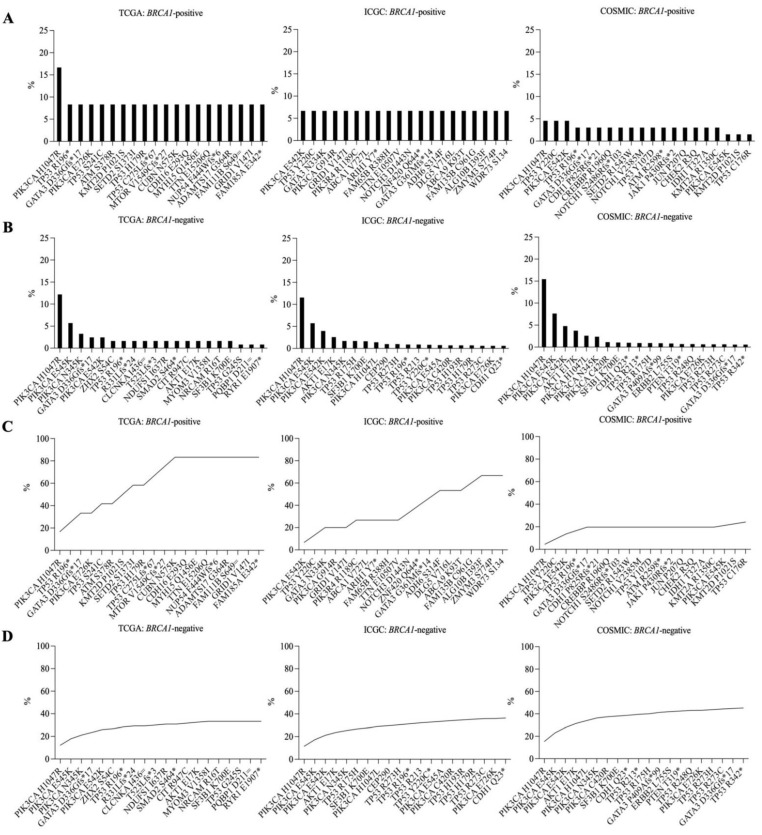
Top recurrent somatic mutations of (**A**) *BRCA1*-positive, and (**B**) *BRCA1*-negative samples. Cumulative coverage of recurrent mutations in (**C**) *BRCA1*-positive and (**D**) *BRCA1*-negative samples. *x*-axes indicate coding-region somatic mutations and the *y*-axes indicate the percentage of all samples with the corresponding mutation. (* indicates stop gained variants).

**Figure 7 vaccines-10-01597-f007:**
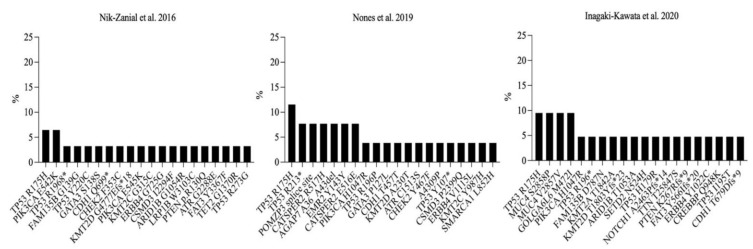
Top recurrent somatic mutations of germline *BRCA1*-mutated breast cancer studies. X-axes indicate coding-region somatic mutations and the Y-axes indicate the percentage of all samples with the corresponding mutation. (* indicates stop gained variants).

**Table 1 vaccines-10-01597-t001:** Characteristics of *BRCA1*-positive samples.

	TCGA	ICGC	COSMIC
Sample size (*n*)	12	15	66
Age			
Mean	59.58 (±12.86)	58.73 (±16.26)	54.14 (±14.91)
Unknown	0	0	38
Sex			
Female	12 (100%)	15 (100%)	65 (98.48%)
Male	0	0	1 (1.52%)
The American Joint Committee on Cancer (AJCC) stage			
I	3 (25.00%)	1 (6.67%)	0
II	8 (66.67%)	9 (60.00%)	3 (4.55%)
III	0	5 (33.33%)	7 (10.61%)
IV	0	0	4 (6.06%)
Unknown	1 (8.33%)	0	52 (78.78%)
Histology type (count/%)			
Infiltrating ductal carcinoma, NOS	11 (91.67%)	14 (93.33%)	20 (30.30%)
Lobular carcinoma, NOS	0	0	3 (4.55%)
Metaplastic carcinoma, NOS	1 (8.33%)	1 (6.67%)	0
Acini cell carcinoma	0	0	1 (1.51%)
Phyllodes tumor	0	0	1 (1.51%)
Unknown	0	0	42 (63.63%)
Molecular subtype (count/%)			
ER –, HER2 –	0	2 (13.33%)	5 (7.58%)
ER –, HER2 +	0	2 (13.33%)	1 (1.51%)
ER +, HER2 –	0	1 (6.67%)	3 (4.55%)
ER +, HER2 +	0	0	0
Hormone receptor +	0	0	29 (43.94%)
Hormone receptor −	0	0	0
Unknown	12 (100%)	10 (66.67%)	28 (42.42%)
Sequencing data type			
WGS	0	7 (46.67%)	11 (16.67%)
WES	12 (100%)	8 (53.33%)	9 (13.64%)
Target sequencing	0	0	46 (69.69%)

**Table 2 vaccines-10-01597-t002:** Characteristics of *BRCA1*-negative samples.

	TCGA	ICGC	COSMIC
Sample size	123	1714	3158
Age			
Mean	58.34 (±13.68)	56.67 (±13.85)	59.39 (±12.49)
Unknown	3	116	2953
Sex			
Female	119 (96.74%)	1699 (99.12%)	2989 (94.65%)
Male	4 (3.25%)	15 (0.87%)	169 (5.35%)
The American Joint Committee on Cancer (AJCC) stage			
I	16 (13.00%)	51 (2.97%)	314 (9.94%)
II	65 (52.84%)	96 (5.60%)	257 (8.14%)
III	36 (29.26%)	31 (1.80%)	158 (5.00%)
IV	4 (3.25%)	4 (0.23%)	185 (5.86%)
Unknown	2 (1.62%)	1532 (89.38%)	2244 (71.06%)
Histology type (count/%)			
Infiltrating ductal carcinoma, NOS	102 (82.92%)	485 (28.29%)	1304 (41.29%)
Lobular carcinoma, NOS	15 (12.19%)	36 (2.10%)	258 (8.17%)
Infiltrating ductal and lobular carcinoma	2 (1.62%)	0	76 (2.41%)
Tubular carcinoma	0	6 (0.35%)	0
Metaplastic carcinoma, NOS	2 (1.62%)	4 (0.23%)	16 (0.51%)
Papillary carcinoma	1 (0.81%)	18 (1.05%)	0
Adenocarcinoma, NOS	0	6 (0.35%)	0
Mucinous carcinoma	1 (0.81%)	13 (0.75%)	0
Adenoid cystic carcinoma	0	1 (0.05%)	0
Acini cell carcinoma	0	0	13 (0.41%)
Carcinoma with neuroendocrine features	0	1 (0.05%)	0
Unknown	0	1144 (66.74%)	1491 (47.21%)
Molecular subtype (count/%)			
ER−, HER2−	0	0	386 (12.22%)
ER−, HER2+	0	0	52 (1.65%)
ER+, HER2−	0	477 (27.82%)	0
ER+, HER2+	0	0	19 (0.60%)
Hormone receptor+	0	0	1132 (35.85%)
Hormone receptor−	0	0	0
Unknown	123 (100%)	1237 (72.17%)	1569 (49.68%)
Sequencing data type			
WGS	0	679 (49.61%)	70 (2.22%)
WES	123 (100%)	1035 (60.38%)	9 (0.28%)
Target sequencing	0	0	3079 (97.50%)

**Table 3 vaccines-10-01597-t003:** Binding affinity prediction results between peptides of recurrent somatic mutations and MHC class I. The table showed only peptide-MHC class I pairs with a predicted binding affinity of less than 100 nM.

Somatic Mutation	Peptide	Length	IC50	Percentile Rank	HLA Type	Allele Frequency
			(NetMHCpan BA)	(NetMHCpan EL)		(%)
*PIK3CA* H1047R	YFMKQMNDAR	10	51.2	0.51	HLA-A*33:03	3.28
	EYFMKQMNDAR	11	37.03	0.08	HLA-A*33:01	1.9
	EYFMKQMNDAR	11	64.4	0.21	HLA-A*33:03	3.28
	FMKQMNDAR	9	68.14	0.54	HLA-A*33:03	3.28
	YFMKQMNDAR	10	65.48	0.41	HLA-A*33:01	1.9
*PIK3CA* E542K	KITEQEKDFLW	11	73.72	0.18	HLA-B*58:01	3.53
*PIK3CA* E545K	SEITKQEKDFLW	12	47.3	0.06	HLA-B*44:03	3.42
	ITKQEKDFLW	10	87.18	0.35	HLA-B*15:17	1.15
	ITKQEKDFLW	10	14.11	0.01	HLA-B*57:01	2.33
	ITKQEKDFLW	10	83.49	0.06	HLA-B*57:02	0.28
	ITKQEKDFLW	10	29.83	0.03	HLA-B*57:03	0.51
	ITKQEKDFLW	10	49.48	0.02	HLA-B*57:04	0.19
	ITKQEKDFLW	10	13.03	0.04	HLA-B*58:01	3.53
	SEITKQEKDFLW	12	37.97	0.03	HLA-B*44:02	3.76
*PIK3CA* N345K	ATYVKVNIR	9	18.66	0.04	HLA-A*31:01	2.35
	ATYVKVNIR	9	95.92	0.34	HLA-A*68:01	3.2
	CATYVKVNIR	10	49.13	2.3	HLA-A*68:01	3.2
	IKILCATYVK	10	87.01	2.9	HLA-A*03:02	2.33
	IKILCATYVK	10	91.23	3	HLA-A*11:01	12.96
	IKILCATYVK	10	91.23	3	HLA-A*11:02	1.75
	ILCATYVKV	9	49.53	0.7	HLA-A*02:03	3.28
	ILCATYVKV	9	53.62	0.8	HLA-A*02:02	1.87
	ILCATYVKV	9	80.57	0.58	HLA-A*02:01	15.67
	KILCATYVK	9	16.81	0.42	HLA-A*11:01	12.96
	KILCATYVK	9	16.81	0.42	HLA-A*11:02	1.75
	KILCATYVK	9	18.78	0.32	HLA-A*03:02	2.33
	KILCATYVK	9	39.61	0.53	HLA-A*03:01	7.29
	KILCATYVK	9	48.12	0.56	HLA-A*30:01	2.98
	KILCATYVK	9	98.38	2.1	HLA-A*31:01	2.35
	KILCATYVKV	10	83.05	3.1	HLA-A*02:06	1.82
	RIKILCATYVK	11	59.05	0.64	HLA-A*30:01	2.98
*TP53* R175H	-	-	-	-	-	-
*TP53* R196*	-	-	-	-	-	-
*TP53* Y220C	VVPCEPPEV	9	142.34	0.28	HLA-A*02:06	1.82

## Data Availability

The datasets generated during and analyzed during the current study are available from the corresponding author upon reasonable request.

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
