# Peer review of "Identification of Shared Neoantigens in BRCA1-Related Breast Cancer"

_vaccines, 2022, doi:10.3390/vaccines10101597_

Round 1

Reviewer 1 Report

Dear Editor Vaccine

This MS is described " Identification of shared neoantigens in BRCA1-related breast cancer. This is an interesting study and a complete and comprehensive study has been carried out. However, some points need to be addressed before publication.

·       In discussion, the difference between this study and other studies should be explained in more detail.

·       Please explain the conclusion in more detail.

Best Regards,

Author Response

Response to Reviewer 1 comments: 

We really appreciate the reviewer’s valuable responses and suggestions. We are also 
thankful that the reviewer appreciated our work and spent valuable time to commenting 
to improve our work before publication. Here are our responses to the reviewer’s 
comments. 

Reviewer 1: 

This MS is described " Identification of shared neoantigens in BRCA1-related breast 
cancer. This is an interesting study and a complete and comprehensive study has been 
carried out. However, some points need to be addressed before publication. 

1. In discussion, the difference between this study and other studies should be 
explained in more detail. 

We thank the reviewer for this invaluable suggestion. Per the reviewer’s comment, 
we have written an additional detail to highlight the differences between our study 
and the previous works in the introduction section (Line 84-89). We have 
mentioned a similar study that was done in uncategorized/unspecified breast 
cancer samples to provide a contrast to our work. We also revised a paragraph in 
the discussion section to highlight the key findings in this study compared to the 
previous report (Line 522-548). 

2. Please explain the conclusion in more detail. 

We appreciate the reviewer for this valuable comment. Per this comment, we have 
added a conclusion section into the manuscript and carefully summarized our 
findings, included a brief description of future works and potential applications 
(Line 628-641). 

Reviewer 2 Report

Reviewer comments:

Comments to Author

This manuscript by Dr. Ruangapirom and group on “Identification of shared neoantigens in BRCA1-related breast cancer” is highlighting upon the number of potential shared neoantigens among BRCA1-related breast cancer which may be used in developing off-the-shelf neoantigen-based vaccines. The experimental designing is impressive, and the manuscript is for the most part well written with substantial evidence of confirmatory and supplementary data. The discussion is also well goes with the results and postulated according to the evidence provided. The references are appropriate and timely.

Minor criticisms

• Labels on the X-axis should be clearly stated in the Figure 6.

• Please provide details about the number of samples in each group and describe the statistical analysis in the Figure legends.

• Manuscript should be checked for some typo errors.

Author Response

Response to Reviewer 2 comments: 

We are grateful that the reviewer appreciated our work. The reviewer gave invaluable 
advices/comments on the manuscript. We followed the reviewer’s suggestions as seen 
below. 

Reviewer 2: 

Comments to Author 

This manuscript by Dr. Ruangapirom and group on “Identification of shared neoantigens 
in BRCA1-related breast cancer” is highlighting upon the number of potential shared 
neoantigens among BRCA1-related breast cancer which may be used in developing off-
the-shelf neoantigen-based vaccines. The experimental designing is impressive, and 
the manuscript is for the most part well written with substantial evidence of confirmatory 
and supplementary data. The discussion is also well goes with the results and 
postulated according to the evidence provided. The references are appropriate and 
timely. 

Minor criticisms 

1. Labels on the X-axis should be clearly stated in the Figure 6. 

We thank the reviewer for this comment. Per this comment, we fixed the X-axis 
of Figure 6 by adding more space between the labels to make it visually pleasing 
and easier to understand. We also added a sentence describing the X- and the 
Y- axes in the figure legends. 

2. Please provide details about the number of samples in each group and 
describe the statistical analysis in the Figure legends. 

We appreciate this suggestion from the reviewer. We added a figure (Figure 4C) 
to specifically describe our statistical findings and included the statistical 
description in the figure legend as suggested by the reviewer (Line 357-362). 
Statistical analyses were not performed in the other figures; however, we made 
sure that the number of total counts in each analysis were included in all the 
figures. 

3. Manuscript should be checked for some typo errors. 

We appreciate this valuable comment from the reviewer. The manuscript was 
carefully re-checked for typographical and grammatical errors. Corrections were 
made accordingly. 

Reviewer 3 Report

The article entitled ' Identification of shared neoantigens in BRCA1-related breast cancer’ by Ruangapirom et al,  greatly describes the potential neoantigens for therapeutic interventions to treat breast cancer. The article is very interesting and provides evidence.  However, The authors could improve the article.

1. The authors need to describe the number of cases of breast cancer globally, its incidence, occurrence and demography.

2. Can the authors integrate the number of genes identified in BRCA1 positive and negative samples and identify unique antigens that are expressed in both sub types. It would be useful to understand the overlap between these two groups.

3. They need to complement the data obtained with some invitro data from cell lines.

4. The sample size is too small, did they consider these samples specifically from a demographic location. Otherwise they can increase the sample size.

5. The study mainly provides Bioinformatic analysis data. The proposed neoantigens  need to be confirmed by other methods.

6. An outline of the next steps of the process will be more insightful and needs to be included in the conclusion. (such as immunization of mice with these antigens and looking for immunological phenotypes of markers). An illustarative flow chart would be useful for the readers.

7. Review of English language is also required for the article.

Author Response

Responses to Reviewer 3 comments: 

We appreciate the reviewer’s insightful comments. Several important points 
were raised by the reviewer to which we carefully addressed as followed. 

Reviewer 3: 

Comments and Suggestions for Authors 

The article entitled ' Identification of shared neoantigens in BRCA1-related 
breast cancer’ by Ruangapirom et al, greatly describes the potential 
neoantigens for therapeutic interventions to treat breast cancer. The article is 
very interesting and provides evidence. However, the authors could improve 
the article. 

1. The authors need to describe the number of cases of breast 
cancer globally, its incidence, occurrence and demography. 

We thank the reviewer for this invaluable comment. Per the reviewer’s 
suggestion, we added the information on the number of cases, 
incidence and demography of breast cancer globally, in the 
introduction section (Line 58-62). The changes are marked by track 
changes. 

2. Can the authors integrate the number of genes identified in 
BRCA1 positive and negative samples and identify unique 
antigens that are expressed in both sub types. It would be useful 
to understand the overlap between these two groups. 

We thank the reviewer for this valuable comment. According to the 
reviewer’s request to compare the unique antigens between both 
subtypes, we provided an additional Venn diagram figure 
(Supplementary figure 1, please also find the figure at the end of this 
document) to illustrate the overlapping of top 20 somatic mutations 
identified in both groups and their corresponding frequencies 
(Supplementary figure 1 and Line 411-427). 

3. They need to complement the data obtained with some in vitro 
data from cell lines. 

We thank the reviewer for this valuable comment. We agree with the 
reviewer and are fully aware that validating bioinformatic findings with 
in vitro study is absolutely crucial. However, with limited time and 
resources, we feel obligated to perform the in vitro validation studies 
and potentially in vivo study in the mouse model in our future work. To 

address this matter as best as we can, we included a paragraph of the 
immunogenicity of antigens previously reported and validated by in 
vitro studies in the discussion section (Line 549-557). 

4. The sample size is too small, did they consider these samples 
specifically from a demographic location. Otherwise they can 
increase the sample size. 

We appreciate the reviewer’s concern about the sample size. We 
initially included all the breast cancer samples available in the 
database, unspecified of their demographic location. However, to 
determine whether or not the sample carries BRCA1 mutation, one of 
the inclusion criteria requires that the BRCA1 gene sequence is 
reported. From there, we included only samples with pathologic or 
likely-pathologic BRCA1 mutations in BRCA1-positive group, which 
further decreased the sample size. Therefore, we did our best to 
include all the relevant samples that were available through all three 
databases. The small sample size may have been attributed by the low 
incidence of pathogenic or likely-pathogenic BRCA1 mutations. 

5. The study mainly provides Bioinformatic analysis data. The 
proposed neoantigens need to be confirmed by other methods. 

We highly appreciate the reviewer’s suggestion. As mentioned in 
response #3, we are fully aware that these are bioinformatic analyses 
and that the confirmation/validation studies are absolutely necessary. 
Therefore, we plan to include those validation and in vivo experiments 
in our future study. 

6. An outline of the next steps of the process will be more insightful 
and needs to be included in the conclusion. (such as 
immunization of mice with these antigens and looking for 
immunological phenotypes of markers). An illustrative flow chart 
would be useful for the readers. 

We thank the reviewer for this insightful comment. As suggested by the 
reviewer, we added a paragraph to discuss future works in the 
discussion section (Line 617-626). We have also added the conclusion 
section that carefully summarizes our findings, including a brief 
description of the next steps of the process (future works) and potential 
applications (Line 628-641). 

7. Review of English language is also required for the article. 

We thank the reviewer for this valuable comment. In fact, we 
have had our manuscript edited by a professional English editor 

prior to submission; however, we totally agree with the reviewer 
that the English writing can still be improved. Therefore, we 
sought help from our colleague whose English is proficient and 
had her review the manuscript. The English editing can be 
found throughout the manuscript. 

Supplementary figure 1 Overlapping top 20 recurrent somatic mutations identified in BRCA1-positive 
and BRCA1-negative groups. The somatic mutations on the lists and in the tables are the overlapped 
recurrent somatic mutations from both subgroups. Its corresponding frequency (percentage of sample 
harboring the somatic mutation in all BRCA1-positive samples or BRCA1-negative samples) are also 
shown in the table for each database.